# Temporal and Serotypic Dynamics of *Actinobacillus pleuropneumoniae* in South African Porcine Populations: A Retrospective Study from 1985 to 2023

**DOI:** 10.3390/pathogens13070599

**Published:** 2024-07-20

**Authors:** Emmanuel M. Seakamela, Marijke M. Henton, Annelize Jonker, Prudence N. Kayoka-Kabongo, Itumeleng Matle

**Affiliations:** 1Bacteriology Division, Agricultural Research Council, Onderstepoort Veterinary Research, Pretoria 0110, South Africa; 2Vetdiagnostix Veterinary Pathology Services, P.O. Box 13624, Cascades 3202, South Africa; henton@vetdx.co.za; 3Department of Veterinary Tropical Diseases, Faculty of Veterinary Science, University of Pretoria, Pretoria 0110, South Africa; annelize.jonker@up.ac.za; 4Department of Agriculture and Animal Health, College of Agriculture and Environmental Sciences, University of South Africa, Johannesburg 1710, South Africa; kabonpnk@unisa.ac.za

**Keywords:** *A. pleuropneumoniae*, pleuropneumonia, pigs, South Africa, serotypes

## Abstract

*Actinobacillus pleuropneumoniae* is a major bacterial pathogen causing porcine pleuropneumoniae, which is a disease of notable economic impact and high fatality rates among pigs worldwide. It has been reported that 19 distinct serotypes of this bacterium exist. Despite its global prominence, there exists a scarcity of information regarding its prevalence and distribution in South Africa. Thus, this study used laboratory records to investigate the serotype diversity, temporal distribution, and seasonal patterns of *A. pleuropneumoniae* isolated from porcine samples spanning from 1985 to 2023 within South Africa. Data from laboratory registries of 354 cases, obtained from three veterinary laboratories in South Africa, were analyzed. The data were categorized into two-time frames: term 1, covering 1985 to 2001, and term 2, spanning from 2002 to 2023. The dataset identified 11 different serotypes, with serotype 7 being the most prevalent at 22.7% (n = 62), which was followed by serotype 5 at 13.8% (n = 42). The study highlighted variations in the prevalence of serotypes among diseased animals over a 38-year period. Serotypes 3, 5, 7 and 8 were commonly observed during this time, while serotype 4 was absent from 1985 to 2001, and serotypes 1, 6, and 10 were absent from 2002 to 2023. The distribution of serotypes showed a diverse variation in the age of affected animals, clinical manifestation, and seasonal occurrence. Key findings revealed that serotype 7 was the most prevalent across all seasons with the highest occurrence in winter. Additionally, Gauteng province showed the highest prevalence of various serotypes. The information collected during this study will serve as a baseline for future epidemiological studies as well as inform control strategies.

## 1. Introduction

*Actinobacillus pleuropneumoniae* is the etiological agent of porcine pleuropneumonia, which is a respiratory disease of domestic pigs (*Sus scrofa domesticus*) [1]. The disease affects pigs of all ages and causes severe economic losses, due to high mortality, costs for antibiotics and vaccinations, longer rearing periods, and lower feed efficiency as well as condemnations of meat at the abattoirs [1,2]. Infection with *A. pleuropneumoniae* may manifest as peracute, acute, subacute, or chronic disease [1]. Acute cases, which are associated with high mortality, are characterized by hemorrhagic fibrinonecrotic bronchopneumonia, while chronic cases exhibit pulmonary sequestra and pleuritis [1]. *A. pleuropneumoniae* primarily resides on the mucous membranes of the respiratory tract but may also be found in nasal discharges and tonsils depending on the disease stage [1]. It is typically transmitted through direct contact with infected pigs, which may serve as asymptomatic carriers. Although uncommon, aerosols carrying the infection can spread between individual pigs and herds over short distances of 1–2 m [3,4].

Currently, *A. pleuropneumoniae* is divided into two biovars and 19 serotypes based on the nicotinamide adenine dinucleotide (NAD) requirements for growth as well as their capsular antigenic properties, respectively [5,6]. In contrast, Mortensen et al. [7] proposed the existence of approximately 18 serotypes, consolidating serotype 9 and 11 into a single entity named serotype 9/11. They noted that the disparity in the complete capsule polysaccharide loci is minimal, differing by only one amino acid, and both share identical toxin profiles (ApxI, and ApxII). Biovar 1 consists of DNA-dependent strains which include serotypes 1–12 and 15–19, while DNA-independent serotypes 13–14 belong to biovar 2 [6,8,9]. Serotype 5 is further subdivided into subtypes 5a and 5b based on minor structural differences seen within the structure of the polysaccharides within these two serotypes [10,11].

Although *A. pleuropneumoniae* serotypes have been extensively studied in many parts of the developed world [12,13], there is a dearth of information regarding their prevalence and distribution in low–middle-income countries, including South Africa. The common serotypes of *A. pleuropneumoniae* exhibit global variability in terms of geographical distribution, prevalence, and their capacity to affect hosts (virulence) [1]. The geographical distribution of *A. pleuropneumoniae* serotypes varies from country to country [1]. For instance, in Japan, serotype 2 is common [14], while in Canada and the USA, serotypes 5 and 7 prevail [15]. In Europe, serotype 2 prevails in Belgium, Hungary, and Denmark [14,15,16], while serotype 8 dominates in England [17]. New Zealand sees dominance in serotype 7, and China exhibits dominance in serotypes 1, 5, and 7 [18]. The prevalence of predominant serotypes may change over time, often coinciding with the introduction of new strains through various means including livestock movement [19]. These epidemiological data are essential for mapping disease outbreaks, identifying newly introduced serotypes, and assisting in vaccine production research [17].

Principal virulence factors in *A. pleuropneumoniae* include exotoxins, lipopolysaccharides, capsular antigens and outer membrane proteins [20]. Exotoxins, particularly Apx I, Apx II, Apx III and Apx IV are expressed to a varying degree by all the different serotypes of *A. pleuropneumoniae* [21,22]. The lipopolysaccharides (LPS), capsular antigens and the outer membrane proteins act as adhesion factors, phagocytosis evaders and iron transporters, respectively, thereby promoting growth and survival within the host [23,24,25].

Due to its importance as a pathogen from both animal health and economic standpoints, there is a strong incentive to monitor the spread of *A. pleuropneumoniae* serotypes in porcine populations. However, there have been limited studies conducted in South Africa since the first isolation of *A. pleuropneumoniae* in 1985 to establish its epidemiology in the porcine population. Moreover, knowledge of serotype distribution is important for formulating appropriate control strategies. This study aimed to use laboratory records to investigate the serotype diversity, distribution, and seasonal occurrence of *A. pleuropneumoniae* isolated from pig samples during the period 1985 to 2023 in veterinary laboratories in South Africa. The data generated from this study will serve as a baseline for future epidemiological studies and will also inform prevention and control strategies.

## 2. Materials and Methods

### 2.1. Study Design and Study Area

A retrospective and longitudinal analysis of laboratory test records archived at the Agricultural Research Council: Onderstepoort Veterinary Research (ARC: OVR), Idexx, and Vetdiagnostix, South Africa, was used for this study. This methodology was selected due to its cost-effectiveness and efficiency in data collection. These laboratories are crucial in supporting both government and private sectors, as well as farmers, by providing essential laboratory services. The data used in this study were obtained from records of clinical cases involving animals suspected to be suffering from porcine pleuropneumonia between 1985 and 2023. A case was defined as all the samples from one source received in one day. The dataset comprised demographic information (age, year of specimen collection, specimen source, main clinical manifestations, season, and provinces) as well as serotypes 

Owing to the changes in the data capture method at the laboratory post-2001, the analysis was conducted in two distinct phases to enhance the study’s impact. The first period (Term 1) encompasses data collected between 1985 and 2001, while the second period (Term 2) covers the period from 2002 to 2023. The data from the first term are more comprehensive, they include a wealth of demographic information for each sample, some of which was not captured in the second phase (see Figure 1).

### 2.2. Laboratory Analysis Procedures

#### 2.2.1. Isolation and Identification of *A. pleuropneumoniae*

All respiratory tract (lung and tonsils) samples from pigs, regardless of whether pleuropneumonia was suspected or not, were cultured on 5% sheep blood tryptose agar with a *Staphylococcus* feeder streak in a 5–10% carbon dioxide atmosphere for at least 48 h [26]. After 48 h, suspicious single bacterial colonies were subcultured for further typing by their cultural and biochemical properties. Hemolytic, Christie–Atkins–Munch-Peterson (CAMP) positive or NAD-dependent isolates were examined, and those that were CAMP positive, urease positive, and indole negative were serotyped. In cases where the identity was uncertain, additional inoculation with lactose, raffinose, mannitol, and sorbitol was performed [27].

#### 2.2.2. Serotyping

Isolates for serotyping were cultured on Mycoplasma Agar base (PPLO agar) as recommended. The immunological-based serotyping was conducted through a slide agglutination test, employing rabbit antisera against serotypes 1 through 10 and 12. Typing sera for the 11 strains were prepared in rabbits as described [28]. As the prepared antisera for types 6 and 8 gave unacceptably high levels of cross-reactions with each other, these two antisera were cross-absorbed with each other [29]. The ARC: OVR laboratory lacked antisera for serotypes 11, 13 to 19 during this study period. After July 2000, any isolate giving an equivocal result in the rapid test was retested using a boiled extract [29].

### 2.3. Statistical Analysis

The information was captured onto an Excel spreadsheet and analyzed using IBM SPSS v.29 to generate the descriptive statistics such as frequencies and proportions. The frequency and proportion of serotypes were analyzed by dividing the frequency of each serotype by a total number of all serotypes. Pearson’s chi-square (χ2) test was used to test for associations between serotypes and the year of isolation, seasonality, and provinces. A *p*-value ≤ 0.05 was considered statistically significant, and the 95% confidence intervals were determined.

## 3. Results

A dataset comprising 354 cases of positive *A. pleuropneumoniae* isolated between 1985 and 2023 was evaluated for this study. Table 1 shows the distribution of *A. pleuropneumoniae* cases in nine provinces of South Africa with the highest proportion from Gauteng (35.7%; n = 123), Mpumalanga (n = 59), Western Cape (n = 57), Kwa-Zulu Natal (n = 42) and Limpopo province (n = 37). Of the 354 cases, 74.6% (n = 264) indicated that *A. pleuropneumoniae* was isolated as pure cultures from samples, while 25.4% (n = 90) reported its isolation in mixed culture with other bacterial species. The records revealed coinfections involving *Pasteurella multocida* (51.0%), *Streptococcus suis* (17.9%), *Trueperella pyogenes* (17.9%), *Staphylococcus* species (8.0%), *Bordetella bronchiseptica* (2.7%), *Mycoplasma* (1.8%), and *Salmonella* species (0.9%). Furthermore, of the 354 cases, 50 were not subjected to serotyping and therefore were excluded from further analysis.

Overall, 90 farms were affected across the country, with farms from Gauteng, KwaZulu-Natal, Limpopo, and the Western Cape dominating the other provinces. Of the 90 affected farms, 39 farms experienced at least one episode of the disease during the study period. Specifically, in Gauteng, 24 and 17 episodes were registered in two different farms, while two other farms in the same province had five episodes each. Additionally, between two and four episodes were recorded in seven, four and two farms from Gauteng, Free State, and the Northwest province, respectively. In the Western Cape province, 12, 10, 7, 6 and 5 episodes were recorded in five different farms, while three other farms registered either two or three episodes. Four different farms in Mpumalanga province registered a total of 19 episodes, with the highest count being nine followed by four, three, and two episodes in respective farms. KwaZulu-Natal recorded seven and six episodes in two farms, while the other two farms recorded three episodes each. In Limpopo, two farms each recorded eight episodes, while the other three farms recorded either two or three episodes. Finally, in the Eastern Cape, four episodes were recorded from a single farm.

The analysis of the 304 remaining cases from the dataset identified the presence of 11 different serotypes (1–10, and 12) as well as the rough strains (untypable: 14.8%; n = 45) (Table 2). Among these serotypes, the highest proportions were observed for serotypes 7, 5, 3 and 8 at 22.7% (n = 69), 13.8% (n = 42), 11.2% (n = 34) and 10.9% (n = 33), respectively while the lowest were observed in serotypes 9 (3%; n = 9), 4 (1.6%; n = 5) and 10 (1.3%; n = 4), respectively. Serotypes 1, 2, 6 and 12 recorded proportions of 6.3% (n = 19), 4.9% (n = 15), 5.6% (n = 17) and 3.9% (n = 12).

The analysis of the data based on the year of isolation revealed that 72.4% (n = 220) of the serotypes were detected during the period between 1985 and 2001 (Term 1), while 27.6% (n = 84) of serotypes were isolated during the period from 2002 to 2023 (Term 2). Therefore, a statistically significant difference (*p* < 0.001) was observed in the prevalence of *A. pleuropneumoniae* serotypes during the years under investigation.

Of the 11 serotypes, only 7 (serotypes 2, 3, 5, 7, 8, 9, and 12) were detected throughout the study period. Serotype 4 was not detected between 1985 and 2001 while serotypes 1, 6 and 10 were not detected during the period from 2002 to 2023. The highest frequency of occurrence during the period from 1985 to 2001 was observed in serotype 7 (19.5%; n = 43), which was followed by serotypes 3 (13.2%; n = 29), 8 (10.9%; n = 24), and 5 (10.5%; n = 23). For the period from 2002 to 2023, serotypes 7 (31.0%; n = 26), 5 (22.6%; n = 19), and 8 (10.5%; n = 9) were predominant (Figure 2).

After removing 50 cases from the analysis, Figure 3 shows that serotypes were reported in only eight provinces of South Africa. No isolates from Northern Cape were serotyped. Their distribution by province showed that Gauteng had a statistically significant (*p* < 0.001) higher prevalence (35.7%: n = 109) than Western Cape and Mpumalanga at 18.4% (n = 56) each.

All serotypes, except serotype 4, were prevalent in Gauteng, with serotypes 7 (28.4%), 5 (18.3%), and 3 (11.9%) dominating in the province. Serotypes 1 (26.8%), 5 (21.4%), and 7 (12.5%) were dominant in the Western Cape province, while serotypes 4, 9, and 10 were not detected. All serotypes except 1 were found in Mpumalanga province, with 7 (17.9%), 2 (14.3%), and 3 (12.5%) being the most dominant. In the Northwest province, serotype 8 (15.4%) and 12 (15.4%) were the most prevalent, with serotypes 1, 2, 4, 5, and 6 not being detected. In Limpopo province, serotype 7 (47.6%) and 6 (14.3%) were predominant, while serotypes 1, 2, 4, 9 and 10 were not detected. The Free State province was primarily dominated by serotype 8 (47.6%) and 7 (14.3%) with serotypes 1, 2, 4, 5, 9, 10 and 12 not being detected. In Kwa-Zulu Natal, serotype 8 (33.3%), 3 (16.7%), and 5 (16.7%) were dominant, while serotypes 4, 6, 9, 10 and 12 were not detected. Only serotypes 7 and 10 were detected in the Eastern Cape with serotype 7 (75%) being the most dominant.

Seasonal variation in the occurrences of *A. pleuropneumoniae* serotypes was investigated and revealed the highest prevalence recorded in winter (31.9%; n = 97), which was followed by autumn (28.6%; n = 87), spring (22.4%; n = 68), and summer (17.1%; n = 52). Seven different serotypes (2, 3, 5, 6, 7, 8, and 9) were prevalent throughout the year. Serotypes 7 (20.6%), 5 (15.5%), and 3 (14.4%) were dominant in winter, while serotypes 7 (25.3%), 8 (12.6%) and 5 (11.5%) dominated in autumn. Similar dominance (16.2%) was observed in serotypes 5 and 7 during spring, while serotype 7 (30.8%) and 8 (21.2%) dominated the summer season (Figure 4). Statistical significance (*p* = 0.023) was observed in the distribution of serotypes by season.

Among the 304 cases analyzed, only 20.1% (n = 61) included age data for the affected animals. Analysis indicated that pigs aged 4–6 months (porkers) (44.7%; n = 27) and 2–4 months (weaners) (37.7%; n = 23) were predominantly affected compared to those aged 0–2 months (young piglets) (11.5%; n = 7) and over 6 months (6.6%; n = 4) (*p* = 0.47). Regarding serotype prevalence, serotype 7 (18.5%) was predominant in pigs aged 4–6 months, followed by serotypes 5 and 1, each accounting for 14.8%. Among pigs aged 2–4 months, serotypes 7 (30.4%), 5 (21.7%), 1 (17.4%) and 3 (13.0%) were predominant.

Regarding clinical manifestations, the lesions associated with the disease were very variable and, in most cases, rather non-specific. Fibrinous and hemorrhagic lesions as well as necrotic pneumonia were grouped as acute, while lung abscesses as well as chronic pneumonia and pleuritis were grouped under the heading chronic. Of the 304 cases, only 44.4% (n = 135) of cases recorded the clinical signs of the disease as either acute (39.3%; n = 53) or chronic (60.7%; n = 82). The prevalence of serotypes by clinical signs showed that serotype 7 (24.5%), 3 (13.2%), 1 (11.3%), and 8 (11.3%) dominated the acute cases, while the chronic cases were dominated by serotype 7 (20.7%), 3 (13.4%), 6 (12.2%) and 5 (11%).

## 4. Discussion

This study is the first in South Africa to present data on the distribution of *A. pleuropneumoniae*. In the current study, 25.4% of the cases revealed coinfection with other respiratory bacteria such as *P. multocida*, *S. suis*, *T. pyogenes*, *B. bronchiseptica*, *Staphylococcus*, *Mycoplasma* and *Salmonella* species. Similar results have also been obtained in studies conducted in Uganda [30], Canada [31,32], Australia [33], Germany [34] and the USA [31].

Overall, our results showed that *A. pleuropneumoniae* is prevalent across the country with the highest proportions in Gauteng, Mpumalanga, Western Cape, Kwa-Zulu Natal and Limpopo province.

Of the 19 known serotypes of *A. pleuropneumoniae*, 11 serotypes (1–10, and 12) as well as the rough strains (untypable) were reported in this study. The antisera used in this study represented serotypes 1–10 and 12 only. This suggests that the untypable strains detected in this study might belong to serotypes other than those reported here. The duration of the current study was divided into two terms. The data from the first term were more comprehensive, and serotyping was only performed at the ARC: OVR laboratory. Post-2001, additional laboratories also performed serotyping. The introduction of *A. pleuropneumoniae* PCR typing assay shifted the landscape, as it was found to be simpler, cheaper and enabled phenotype-independent characterization of isolates that were untypable by serotyping [35]. Notably, no rough/untypable strains were reported during the 2nd term of this study. The reasons behind this were not explored as they fell outside the scope of the current study; however, they warrant further investigation in the future.

The prevalence of the serotypes reported in the current study varied significantly (*p* = 0.001) between term 1 and 2. Serotypes 3, 5, 7 and 8 were common at various time periods in the last 38 years in the country. However, serotype 4 was not detected between 1985 and 2001, while serotypes 1, 6, and 10 were not detected during 2002 to 2023. Although South Africa lacks data to benchmark the variation in the occurrence of *A. pleuropneumoniae* serotypes over time, similar fluctuations have also been observed in several countries worldwide. For example, it was reported that serotype 1 dominated in the 1990s, while serotypes 5 and 7 were frequently isolated in Canada in 2014 [16]. In contrast, serotypes 8 and 15 are highly prevalent among diseased pigs in the UK, North America, and Australia, respectively [36,37]. The results of the current study will form a baseline for future epidemiological studies as well as inform the control and prevention strategies for *A. pleuropneumoniae.*

There was a significant difference in the distribution of serotypes in the provinces. Various serotypes dominated in various provinces with serotype 7 frequently isolated in most of the provinces (62.5%; n = 5). The variation in serotype distribution can be attributed to the density of pig farms within each location. This study revealed that Gauteng, Mpumalanga, Western Cape, KwaZulu-Natal and Limpopo provinces reported the highest number of cases and exhibited a diverse range of serotypes. This trend aligns with the findings of the Department of Agriculture, Land Reform and Rural Development (DALRRD) report from 2022 on the South African pork market value chain. According to the report, Limpopo has the highest number of pig farms, accounting for 24%, which was followed by Northwest at 20%. Western Cape, KwaZulu-Natal, and Gauteng each constitute 11%, while Mpumalanga stands at 8%. Northern Cape has the lowest percentage of pig farms at 1%. These statistics suggest that provinces with a higher concentration of pig farms are likely to exhibit a more diverse circulation of serotypes, which was a phenomenon confirmed by our study. The prevalence of 11 serotypes and their dominance in different provinces and pig production farms of South Africa pose a difficulty in the production of serotype-based vaccine. However, vaccines based on APX toxins I–III, which are produced by all *A. pleuropneumoniae* serotypes, offer better protective immunity, especially when combined with other antigens such as Outer Membrane Proteins (OMPs) and lipopolysaccharides [7]. Therefore, it is essential to strengthen farm biosecurity to ensure that dominant serotypes do not spread to other locations and cause outbreaks.

Porcine pleuropneumoniae like any other respiratory infection is associated with cold weather conditions. Our study reported *A. pleuropneumoniae* prevalence throughout the year with the highest in winter, which was expected, and was followed by autumn, spring and summer. According to previous literature, ambient temperatures influence the behavioral activities of pigs [38]. It is reported that during warm temperatures, pigs cool themselves by wallowing in mud [39]. However, pig production housing structures do not cater for such behavior, and therefore, pigs resort to lying on cold floors in solitude, avoiding contact with other individual pigs [39]. This behavior is contrary during cold weather conditions, where pigs huddle [36,37], thereby increasing the chances of transmission, especially in intensive production settings. Furthermore, the invasion of *A. pleuropneumoniae* depends on the inhalation of sufficient bacterial particles to the alveoli, which is prevented by the mucociliary apparatus. During extreme cold temperatures or chilling, the mucociliary apparatus becomes damaged and allows the easy flow of bacterial particles into the alveoli [1]. These might be the reasons for more cases of the disease in the winter season.

In a laboratory-controlled experiment, Assavacheep and Rycroft [40] demonstrated that *A. pleuropneumoniae* can only survive for up to 4 days outside of a pig. However, a study conducted in Mexico isolated *A. pleuropneumoniae* from drinking water and reported its survival for at least 3 weeks in water [41]. The capability of *A. pleuropneumoniae* to form biofilms has been demonstrated in several studies. Other studies have also established that subclinically infected animals carry the bacteria for long time periods until triggered by either infectious or non-infectious factors. Our study also reported the growth of *A. pleuropneumoniae* together with other bacteria. These reasons explain the consistency of *A. pleuropneumoniae* throughout the year in our study. Eight serotypes were consistent across all seasons with serotype 7 being the most frequently detected. The results of our study correlate with the results of the study conducted in Korea, which reported the prevalence of *A. pleuropneumoniae* throughout the seasons [17]. The seasonal variations of this study may inform the planning of the control strategies as well as the understanding of the disease.

*A. pleuropneumoniae* affect pigs of all ages with clinical signs mainly seen in 12–16-week-old pigs [2,42]. In the current study, *A. pleuropneumoniae* serotypes were detected in all age categories (0–2 months; 2–4 months; 4–6 months and over 6 months) with high infection rates in pigs aged 2–6 months, which was as expected. According to previous studies, young piglets may be protected from the infection through antibodies acquired from colostrum during suckling; hence, fewer cases are reported (11.7%). However, low immunity due to the decreased uptake of antibodies and the duration of exposure may render piglets susceptible [1,43,44,45], which might have been the reason for the mortality of pigs aged 0–2 months. According to Chiers et al., [46], regardless of the maternal antibodies, *A. pleuropneumoniae* can colonize the upper respiratory tract. These results highlight that monitoring the level of immunity and proper animal husbandry is key to controlling the disease in the farm.

In terms of clinical manifestations, our study reported higher chronic cases than acute with serotype 7 being consistent in both phases of the disease. The acute phase is associated with high mortality, while the chronic phase is characterized by growth retardation and feed efficiency. Both the acute and chronic phases contribute to financial losses due to expenses incurred for medication, immunization and prolonged feeding [1,46].

## 5. Conclusions

Porcine pleuropneumonia is a globally economically important respiratory disease of pigs. The study established that *A. pleuropneumoniae* is prevalent in all nine pig producing provinces of South Africa. The study has also established that serotype 7 is the commonly isolated serotype in South Africa. Therefore, the authors recommend that more studies, particularly longitudinal, should be conducted in the country to monitor emerging serotypes and the evaluation of vaccine efficacy against the most prevalent serotypes to improve the understanding of the epidemiology of this disease.

## Figures and Tables

**Figure 1 pathogens-13-00599-f001:**
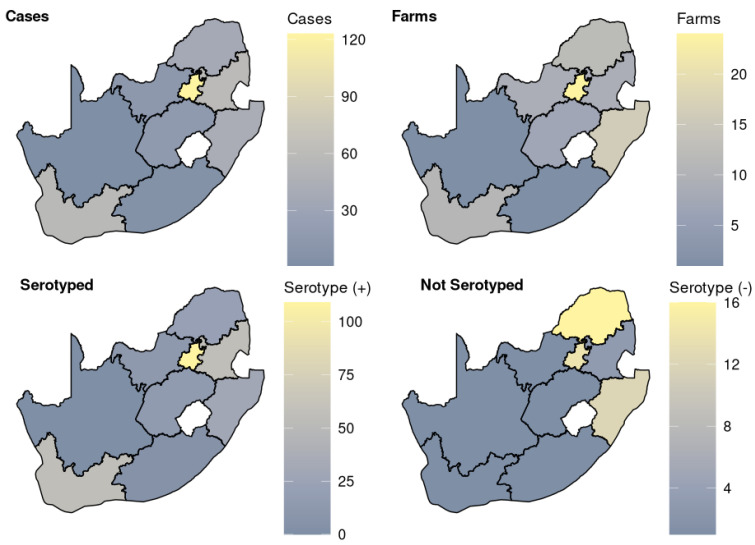
South African map showing the distribution of *Actinobacillus pleuropneumoniae* cases within the provinces.

**Figure 2 pathogens-13-00599-f002:**
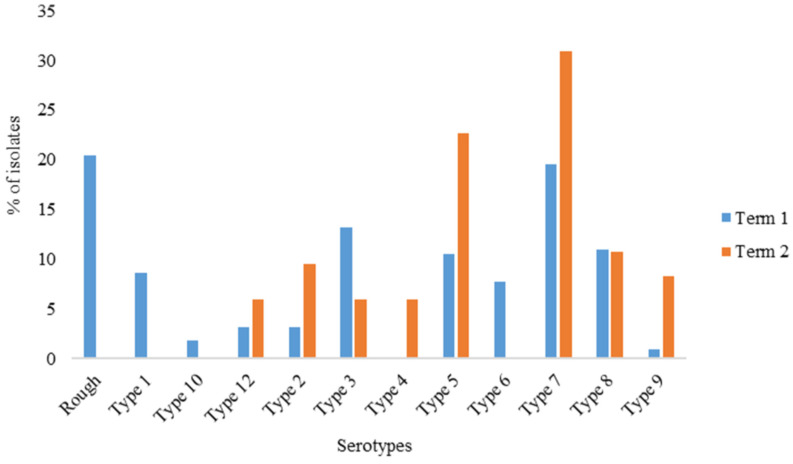
Prevalence of *Actinobacillus pleuropneumoniae* serotypes by term of isolation. Term 1: Year 1985–2001; Term 2: Year 2002–2023.

**Figure 3 pathogens-13-00599-f003:**
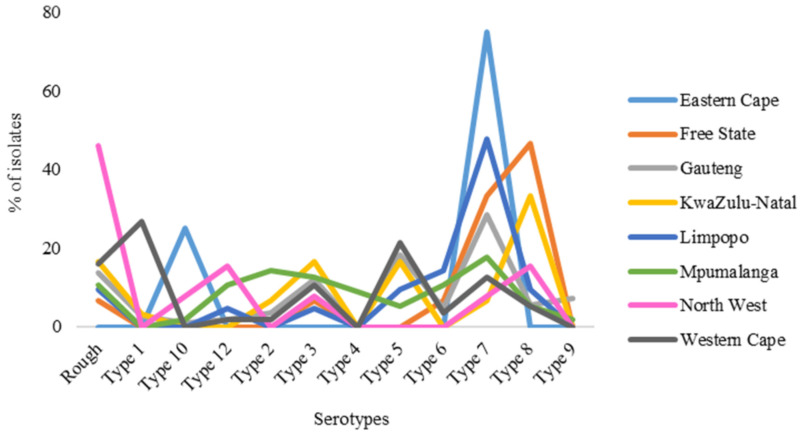
Prevalence of *Actinobacillus pleuropneumoniae* serotypes by location.

**Figure 4 pathogens-13-00599-f004:**
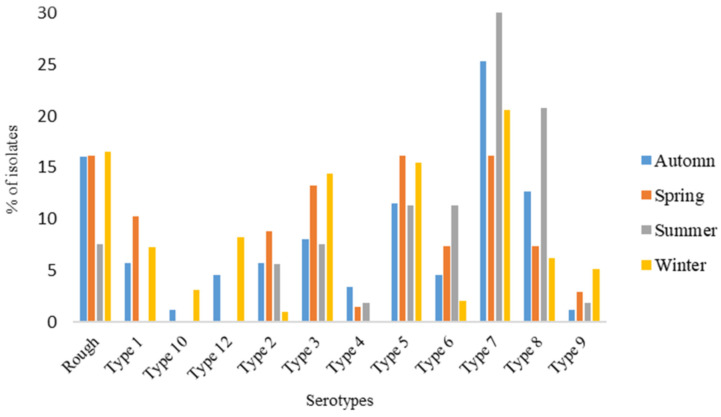
Prevalence of *Actinobacillus pleuropneumoniae* serotypes by season.

**Table 1 pathogens-13-00599-t001:** Distribution of *Actinobacillus pleuropneumonia* cases.

Province	No. of Cases	No of Farms	Serotyped	Not Serotyped
Eastern Cape	5	1	4	1
Free State	16	7	15	1
Gauteng	123	24	109	14
Kwa-Zulu Natal	42	16	30	12
Limpopo	37	12	21	16
Mpumalanga	59	9	56	3
Northwest	14	9	13	1
Western Cape	57	11	56	1
Northern Cape	1	1	0	1
Total	354	90	304	50

**Table 2 pathogens-13-00599-t002:** Prevalence of *Actinobacillus pleuropneumoniae* serotypes.

Serotype	No. of Isolates	95% CI
Rough	45	11–19
Type 1	19	4–9
Type 2	15	3–8
Type 3	34	8–15
Type 4	5	1–4
Type 5	42	10–18
Type 6	17	3–9
Type 7	69	18–28
Type 8	33	8–15
Type 9	9	2–5
Type 10	4	0–3
Type 12	12	2–7
Total	304	99–100

## Data Availability

Data available on request due to privacy/ethical restrictions.

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
