# Peer review of "Temporal and Serotypic Dynamics of Actinobacillus pleuropneumoniae in South African Porcine Populations: A Retrospective Study from 1985 to 2023"

_pathogens, 2024, doi:10.3390/pathogens13070599_

Round 1

Reviewer 1 Report

Comments and Suggestions for Authors

General comments

  • To make the text consistent, either use the full form or abbreviations of  Actinobacillus pleuropneumonia. For example, line 39.
  1. Abstract:
    • Add a sentence summarising the key findings related to the seasonal and geographical distribution of the serotypes. For example, "Key findings revealed that serotype 7 was the most prevalent across all seasons, with the highest occurrence in winter, and Gauteng province showed the highest prevalence of various serotypes."
  2. Introduction:
    • Expand on the impact of A. pleuropneumoniae by discussing its economic and animal welfare implications on the global swine industry.
    • Line 63. The sentence needs revision.
    • Line 66 needs reference.
    • Line 76: is it to mean “..animal health and economic stand-76 point”?
  3. Materials and Methods:
    • Explain how the challenges faced with cross-reactivity in serotyping, particularly between serotypes 6 and 8, are managed, and describe the measures taken to ensure accurate serotyping.
    •  Have you tried to carry out additional statistical analyses, such as logistic regression or multivariate analysis, to explore potential confounding factors that might influence the distribution of serotypes, such as farm size, biosecurity measures, and pig density?
    • Line 92-93: what is the importance of this sentence to the study?
  4. Results:
    • Based on the title and introduction, this study has placed great emphasis on the temporal and serotype distribution, yet no maps have been presented. Add graphs that illustrate the temporal trends of serotype prevalence over the study period. For example, a line graph showing the prevalence of the top serotypes over time can visually highlight trends and changes.
    • Consider adding geographical maps showing the distribution of serotypes across different provinces for a clearer spatial understanding of the data.
    • Lines 146-156: you can add the episode information to Table 1.
    • Ensure all figures and tables are self-explanatory with detailed legends. For example, in Figures 1 and 2, A. pleuropneumonia is indicated without describing its abbreviation. Please be consistent with the use of words.
  5. Discussion:
    • Generally, the discussion part lacks coherence and flow and is hard to follow. The discussion should focus more on interpreting the results rather than reiterating them.
    • Discuss the potential reasons for seasonal variations in more detail. For example, explore environmental factors, farm management practices, and pig housing conditions that might contribute to higher prevalence in winter.
    • Try to add some explanation for how the dominance of certain serotypes might influence the formulation of vaccines and the focus of biosecurity measures.
  6. Conclusion:
    • Line 337 isn't necessary here, as you can't make global conclusions based on a single country’s data. Remove it.
    • Strengthen the conclusion by providing specific recommendations for future research, such as the need for longitudinal studies to monitor emerging serotypes and the evaluation of vaccine efficacy against the most prevalent serotypes ( if any).

Comments on the Quality of English Language

Refer to the above comment.

Author Response

Reviewer 1

  • To make the text consistent, either use the full form or abbreviations of Actinobacillus pleuropneumonia. For example, line 39.
  • Response: Thank you for this comment, the authors have applied the abbreviation as suggested throughout the document.
  1. Abstract:
    • Add a sentence summarising the key findings related to the seasonal and geographical distribution of the serotypes. For example, "Key findings revealed that serotype 7 was the most prevalent across all seasons, with the highest occurrence in winter, and Gauteng province showed the highest prevalence of various serotypes."
  • Response: Thank you for making this suggestion, the authors have made adjustment as suggested by the reviewer. Line 28-30 “Key findings revealed that serotype 7 was the most prevalent across all seasons, with the highest occurrence in winter. Additionally, the Gauteng province exhibited the highest prevalence of various serotypes. Line
  1. Introduction:
    • Expand on the impact of A. pleuropneumoniae by discussing its economic and animal welfare implications on the global swine industry.
    • Response: Thank you for this comment, the authors feel strong that this comment will improve the content of the manuscript as such we have added “cost for antimicrobials and vaccinations” and “condemnations of meat at the abattoirs” to the sentence. Line 37 - 40
    • Line 63. The sentence needs revision.
    • Response: Thank you. The authors have revised the sentence accordingly and it now reads as “The geographical distribution of A. pleuropneumoniae serotypes varies from country to country. Line 63 - 64
    • Line 66 needs reference.
    • Response: Reference [20] has been added
    • Line 76: is it to mean “..animal health and economic stand-76 point”?
    • Response: Agreed, we have removed “an” before “animal” Line 79
  2. Materials and Methods:
    • Explain how the challenges faced with cross-reactivity in serotyping, particularly between serotypes 6 and 8, are managed, and describe the measures taken to ensure accurate serotyping.
    • Response: Thank you for the comment. The authors have added a reference to the method Line 125 - 132
    •  Have you tried to carry out additional statistical analyses, such as logistic regression or multivariate analysis, to explore potential confounding factors that might influence the distribution of serotypes, such as farm size, biosecurity measures, and pig density?
    • Response: Thank you for this comment. The authors did consider conducting additional statistical analyses as suggested. However, the nature of this data made it extremely difficult to do so. Primarily, our data included only positive cases that were retrospectively captured. Secondly, the available data did not include information such as farm size, biosecurity practices, and pig density. Finally, some data points have changed over the years, and the authors could not go back to obtain the missing information for further analyses.
    • Line 92-93: what is the importance of this sentence to the study?
    • Response: the sentence has been modified Line 91 - 93
  3. Results:
    • Based on the title and introduction, this study has placed great emphasis on the temporal and serotype distribution, yet no maps have been presented. Add graphs that illustrate the temporal trends of serotype prevalence over the study period. For example, a line graph showing the prevalence of the top serotypes over time can visually highlight trends and changes.
    • Response: The study assessed data spanning 38 years with eleven serotypes discovered. The graph indicating 38 years with 11 serotypes per year was not clear, hence we categorised the years into term 1 and term 2 for clear visibility and understanding.  
    • Consider adding geographical maps showing the distribution of serotypes across different provinces for a clearer spatial understanding of the data.
    • Response: Thank you for the comment. The authors have inserted the map.
    • Lines 146-156: you can add the episode information to Table 1.
    • Response: Cases shown in table 1 are also a representative of episodes
    • Ensure all figures and tables are self-explanatory with detailed legends. For example, in Figures 1 and 2, A. pleuropneumonia is indicated without describing its abbreviation. Please be consistent with the use of words.
    • Response: Corrected
  4. Discussion:
    • Generally, the discussion part lacks coherence and flow and is hard to follow. The discussion should focus more on interpreting the results rather than reiterating them.

Response: Thank you for the comment. The authors have revised the discussion

    • Discuss the potential reasons for seasonal variations in more detail. For example, explore environmental factors, farm management practices, and pig housing conditions that might contribute to higher prevalence in winter.
    • Response: Thank you for the comment. The authors have added a paragraph on that Line 299 - 310
    • Try to add some explanation for how the dominance of certain serotypes might influence the formulation of vaccines and the focus of biosecurity measures.
    • Response: Thank you for the comment. The authors have added a paragraph on that. Line 289 - 295
  1. Conclusion:
    • Line 337 isn't necessary here, as you can't make global conclusions based on a single country’s data. Remove it.
    • Response: sentences removed as suggested
    • Strengthen the conclusion by providing specific recommendations for future research, such as the need for longitudinal studies to monitor emerging serotypes and the evaluation of vaccine efficacy against the most prevalent serotypes ( if any).
    • Response: Agreed, we added the following sentence as recommendations:  Therefore, the authors recommend that more studies particularly longitudinal, should be conducted in the country to monitor emerging serotypes and the evaluation of vaccine efficacy against the most prevalent serotypes so as to improve the understanding of the epidemiology of this disease. Line 347 - 348

Reviewer 2 Report

Comments and Suggestions for Authors

1.      Lines 33-44: Consider providing references after each sentence. Besides, there is a newer Disease of Swine, available.

2.      Line 46: The number of references seems excessive.

3.      Line 47: consider moving this reference to the end of the sentence.

4.      Choose either 18/19 or eighteen/nineteen

5.      Lines 70-75: this information seems irrelevant to the topic of the study. Consider removing.

6.      I think the manuscript will only benefit if you include some studies on the prevalence/detection of the pathogen in Africa.

7.      Lines 94-96: more comprehensive data on the clinical status of pigs used in this study would be helpful to link detection of the pathogen with certain pathologies.

8.      Line 106: please italicize.

9.      Line 107: please specify what kind of samples were collected.

10. Lines 107-113: please double check if the reference 30 can be used for the whole.

11. Line 115: similarly, please provide a reference for this method of culturing.

12. Lines 120-121: does it mean that isolated strains were not tested for these serotypes?

13. Line 131: Please provide some information on characteristics of the farms used to collect samples (size).

14. Lines 137-139: If you want to report detection of other pathogens, consider providing all the methodology.

15. Lines 139-140: please clarify why these samples were excluded from serotyping.

16. Lines 143-156: a clarification on how these farms were chosen to be tested would be beneficial for readers. What was considered as a case? How many samples were collected from each case? Respiratory pathology is very common so the statement that 51 farms never experienced pathology where APP could have been involved in seems unrealistic.

17. Lines 165-169. In order to confirm the prevalence of the pathogen between the two time periods, it is important to have a similar number of samples collected. Besides, it is not clear whether the same farms/regions were studied for both terms.

18. Line 182: please provide a method of how the p value was calculated. Besides, a reader may expect to see results for all 354/304 cases.

19. In order to evaluate the age-specific prevalence of the pathogen, cross-sectional or longitudinal approaches should be used.

Author Response

Reviewer 2 Suggestions for Authors

  1. Lines 33-44: Consider providing references after each sentence. Besides, there is a newer Disease of Swine, available.

Response: References added Line 115 - 123

  1. Line 46: The number of references seems excessive.

Response: References have been minimised Line 50

  1. Line 47: consider moving this reference to the end of the sentence.

Response: we have moved reference [6,9] to the end of the sentence Line 50

  1. Choose either 18/19 or eighteen/nineteen

Response: We have changed “nineteen” to “19” Line 48

  1. Lines 70-75: this information seems irrelevant to the topic of the study. Consider removing.

Response: Agreed, We, have removed the sentence

  1. I think the manuscript will only benefit if you include some studies on the prevalence/detection of the pathogen in Africa.

Response: We agree, however the scientific publications on the disease are scarce in Africa

  1. Lines 94-96: more comprehensive data on the clinical status of pigs used in this study would be helpful to link detection of the pathogen with certain pathologies.

Response: Agreed. The clinical manifestations of the disease have been added. Line 226 - 229

  1. Line 106: please italicize.

Response: I have italicised “A. pleuropneumonia”. Line 114

  1. Line 107: please specify what kind of samples were collected.

Response: Majority of samples brought into the laboratory were lung samples while a small percentage was tonsils. For this reason, sample type was not included as a variable Line 115

  1. Lines 107-113: please double check if the reference 30 can be used for the whole.

Response: Another reference [27] has been added Line 118

  1. Line 115: similarly, please provide a reference for this method of culturing.

Response: Reference added Line 115 - 123

  1. Lines 120-121: does it mean that isolated strains were not tested for these serotypes?

Response: This study was based on retrospective data, Yes the isolates were tested for only 11 serotypes

  1. Line 131: Please provide some information on characteristics of the farms used to collect samples (size).

Response: The samples were collected and sent to the laboratory by the attending veterinarian. Most of the farms included in this study are commercial farms.

  1. Lines 137-139: If you want to report detection of other pathogens, consider providing all the methodology.

Response: Agreed, we have added a sentence to the isolation of bacteria in methods Line 118 - 119

  1. Lines 139-140: please clarify why these samples were excluded from serotyping.

Response: Please note that the study used retrospective data. Therefore, serotyping was only done on request from the client which suggest that serotyping was not requested  

  1. Lines 143-156: a clarification on how these farms were chosen to be tested would be beneficial for readers. What was considered as a case? How many samples were collected from each case? Respiratory pathology is very common so the statement that 51 farms never experienced pathology where APP could have been involved in seems unrealistic.

Response: This study is based on retrospective data of samples brought to the laboratories for routine diagnosis. All farms mentioned in this study had experienced an episode at some point. However, some had more than others’ Line 98 - 99. A case was defined as all the samples from one source received on one day

  1. Lines 165-169. In order to confirm the prevalence of the pathogen between the two time periods, it is important to have a similar number of samples collected. Besides, it is not clear whether the same farms/regions were studied for both terms.

Response: This is a retrospective based study. The period was divided for the purpose of making the data easier to analyse.

  1. Line 182: please provide a method of how the p value was calculated. Besides, a reader may expect to see results for all 354/304 cases.

Response: We have revised the sentence in the method. A p-value ≤ 0.05 was considered statistically significant and the 95% confidence intervals Line 137 - 143

  1. In order to evaluate the age-specific prevalence of the pathogen, cross-sectional or longitudinal approaches should be used.

Response: Agreed and we will put it as one of the recommendations of our study Line 384 - 386

Round 2

Reviewer 1 Report

Comments and Suggestions for Authors

The authors have improved the manuscript; however, several formatting issues persist. For example, figures and tables should be self-explanatory, yet there are still figure legends that use "A. pleuropneumoniae"  without a prior definition of its full form. Additionally, the content in lines 236-244 of the discussion appears irrelevant. The remainder of the manuscript is fine.

Author Response

The authors have improved the manuscript; however, several formatting issues persist. For example, figures and tables should be self-explanatory, yet there are still figure legends that use "A. pleuropneumoniae"  without a prior definition of its full form. Additionally, the content in lines 236-244 of the discussion appears irrelevant. The remainder of the manuscript is fine.

Response: Thank you for the comment. The authors have amended all figures and tables as well as the discussion. Line 239

Reviewer 2 Report

Comments and Suggestions for Authors

This is one of the manuscripts which could be mentioned in your paperDione M, Masembe C, Akol J, Amia W, Kungu J, Lee HS, Wieland B. The importance of on-farm biosecurity: Sero-prevalence and risk factors of bacterial and viral pathogens in smallholder pig systems in Uganda. Acta Trop. 2018 Nov;187:214-221. doi: 10.1016/j.actatropica.2018.06.025. Epub 2018 Jun 24. PMID: 29949731.

Author Response

This is one of the manuscripts which could be mentioned in your paperDione M, Masembe C, Akol J, Amia W, Kungu J, Lee HS, Wieland B. The importance of on-farm biosecurity: Sero-prevalence and risk factors of bacterial and viral pathogens in smallholder pig systems in Uganda. Acta Trop. 2018 Nov;187:214-221. doi: 10.1016/j.actatropica.2018.06.025. Epub 2018 Jun 24. PMID: 29949731.

Response: Thank you for the comment. The authors have added the reference as suggested. Line 243